

# Anvi'o: an advanced analysis and visualization platform for 'omics data

A. Murat Eren[1,2], Özcan C. Esen[1], Christopher Quince[3],
Joseph H. Vineis[1], Hilary G. Morrison[1], Mitchell L. Sogin[1] and
Tom O. Delmont[1]

[1] Josephine Bay Paul Center, Marine Biological Laboratory, Woods Hole, MA, United States
[2] Department of Medicine, The University of Chicago, Chicago, IL, United States
[3] Warwick Medical School, University of Warwick, Coventry, United Kingdom

## ABSTRACT

Advances in high-throughput sequencing and 'omics technologies are revolutionizing studies of naturally occurring microbial communities. Comprehensive investigations of microbial lifestyles require the ability to interactively organize and visualize genetic information and to incorporate subtle differences that enable greater resolution of complex data. Here we introduce anvi'o, an advanced analysis and visualization platform that offers automated and human-guided characterization of microbial genomes in metagenomic assemblies, with interactive interfaces that can link 'omics data from multiple sources into a single, intuitive display. Its extensible visualization approach distills multiple dimensions of information about each contig, offering a dynamic and unified work environment for data exploration, manipulation, and reporting. Using anvi'o, we re-analyzed publicly available datasets and explored temporal genomic changes within naturally occurring microbial populations through *de novo* characterization of single nucleotide variations, and linked cultivar and single-cell genomes with metagenomic and metatranscriptomic data. Anvi'o is an open-source platform that empowers researchers without extensive bioinformatics skills to perform and communicate in-depth analyses on large 'omics datasets.

## INTRODUCTION

High-throughput sequencing of the environmental DNA has become one of the most effective ways to study naturally occurring microbial communities. By circumventing the need for cultivation, shotgun metagenomics—the direct extraction and sequencing of DNA fragments from a sample—provides access to the enormous pool of microbial diversity that marker gene surveys have unveiled (*Handelsman et al., 1998*; *Sogin et al., 2006*). Early studies using capillary sequencing techniques (*Venter et al., 2004*) and, more recently, massively-parallel techniques (*Angly et al., 2006*; *Edwards et al., 2006*), led to descriptions of microbially-mediated activities and their functional interactions that have provided novel insights into medicine (*Turnbaugh et al., 2006*), biotechnology (*Lorenz & Eck, 2005*), and evolution (*Woyke et al., 2006*).

Corresponding author
A. Murat Eren,
a.murat.eren@gmail.com

Current high-throughput sequencing technologies generate an astonishing amount of sequence data, although the lengths of highly accurate DNA sequence reads fall short of bacterial genome sizes by orders of magnitude. Multiple online resources can annotate metagenomic short reads (*Meyer et al., 2008*; *Zakrzewski et al., 2013*), however, their relatively small information content compared to the length of coding regions constrains accurate functional inferences (*Wommack, Bhavsar & Ravel, 2008*; *Carr & Borenstein, 2014*). Despite these limitations, researchers have used metagenomic short reads successfully to investigate and compare the functional potential of various environments (*Tringe et al., 2005*; *Dinsdale et al., 2008*; *Delmont, Simonet & Vogel, 2012*).

The assembly of short reads into contiguous DNA segments (contigs) leads to improved annotations because of the greater information content of longer sequences, including the genomic context of multiple coding regions. Several factors affect the assembly performance (*Pop, 2009*; *Luo et al., 2012*; *Mende et al., 2012*), and the feasibility of the assembly-based approaches varies across environments (*Sharon et al., 2013*; *Iverson et al., 2012*). Nevertheless, increasing read lengths (*Sharon et al., 2015*), novel experimental approaches (*Delmont et al., 2015*), advances in computational tools (*Brown et al., 2012*), and improvements in assembly algorithms and pipelines (*Boisvert et al., 2012*; *Peng et al., 2012*; *Zerbino & Birney, 2008*; *Treangen et al., 2013*) continue to make assembly-based metagenomic workflows more tractable. Additional advances emerge from genomic binning techniques that employ contextual information to organize unconnected contigs into biologically relevant units, i.e., draft genomes, plasmids, and phages (*Venter et al., 2004*; *Tyson et al., 2004*). Draft genomes frequently provide deeper insights into bacterial lifestyles that would otherwise remain unknown (*Stein et al., 1996*; *Alonso-Sáez et al., 2012*; *Kantor et al., 2015*) and offer an opportunity to identify single-nucleotide polymorphisms that differentiate members or strains of a microbial population (*Tyson et al., 2004*). Genome binning processes typically take advantage of sequence composition and the coverage of contigs across multiple samples. Despite associated challenges (*Wooley, Godzik & Friedberg, 2010*; *Luo et al., 2012*), researchers have successfully employed these assembly and binning techniques to identify near-complete novel draft genomes from metagenomic datasets generated from various environments (*Venter et al., 2004*; *Tyson et al., 2004*; *Hess et al., 2011*; *Raveh-Sadka et al., 2015*). This workflow has become more practicable thanks to recently introduced human-guided (*Albertsen et al., 2013*; *Sharon et al., 2013*) and automated (*Alneberg et al., 2014*; *Wu et al., 2014*; *Kang et al., 2015*) approaches and software pipelines that lend themselves to the identification of genome bins.

Beyond these advances, comprehensive analysis of assembled metagenomic data requires the ability to manipulate and mine complex datasets within a visualization framework that immediately reports the end result of these operations. Available tools for the visualization of metagenomic contigs usually employ self-organizing maps (*Sharon et al., 2013*) or principal component analysis plots (*Alneberg et al., 2014*; *Cantor et al., 2015*; *Laczny et al., 2015*). Although these visualization strategies can describe the organization of contigs, they do not present the distribution of contigs across samples along with supporting data such as GC-content, inferred taxonomy, or other automatically generated

or user-specified information for each contig in one display. Interactive visualization tools that report the influence of contextual information on the human-guided contig binning and that provide the ability to modify the membership of contigs in genome bins would improve the quality of draft genomes. A platform that consolidates advanced visualization and analysis infrastructure with an open design that allows the addition of novel algorithms could serve as a test bed for sharing new analytical paradigms and contribute to the dissemination of good practices in the field of metagenomics.

Here we introduce *anvi'o*, an advanced analysis and visualization platform for 'omics data, and describe its assembly-based metagenomic workflow, which includes human-guided and automated metagenomic binning, interactive data exploration, manipulation, visualization, and reporting. To demonstrate anvi'o, we re-analyzed (1) a relatively low-complexity metagenomic dataset from an infant gut microbiome sampled daily (*Sharon et al., 2013*) and (2) a collection of datasets that represent the combined efforts of multiple investigators (*Rodriguez-R et al., 2015*; *Overholt et al., 2013*; *Mason et al., 2012*; *Mason et al., 2014*; *Yergeau et al., 2015*) who studied the microbial response to the 2010 Deepwater Horizon (DWH) oil spill (*Atlas & Hazen, 2011*).

## MATERIAL AND METHODS

Anvi'o is an analysis and visualization platform for 'omics data. It provides an interactive and extensible visualization interface that distills multiple dimensions of information into a single, intuitive display. The platform is written predominantly in Python, JavaScript, and C, and relies on scalable vector graphics (SVG) for most visualization tasks. The visualization core, implemented from scratch in JavaScript, uses low-level SVG object manipulation functions with minimal overhead to optimize performance. Anvi'o displays tree structures with data or metadata layers that describe the properties of each leaf on the tree. The platform stores computed data in self-contained database files that can be interrogated using structured query language (SQL) through SQLite, an open source transactional SQL database engine that does not require any database server or configuration. The user interacts with anvi'o through command line clients or a graphical web browser. The platform generates static HTML web pages to summarize analysis results. Reliance on self-contained database files and static HTML output facilitates transfer of intermediate or final stages of analyses between computers. In this study we emphasize anvi'o's metagenomic workflow, but the platform can also meet the analysis and visualization requirements of other 'omics data types. Anvi'o is a community-driven, open-source project. The source code is licensed under the GNU General Public License, and publicly available at http://merenlab.org/projects/anvio.

### Anvi'o metagenomics workflow

Preparing a metagenomic dataset for an analysis with anvi'o requires a co-assembly of short reads from all or a subset of samples to create community contigs, followed by the mapping of short reads from individual samples back to these contigs. The FASTA file of community contigs and BAM files reporting mapping results for each sample provide the initial input for anvi'o. The BAM file format is the binary representation of the Sequence

Alignment/Map (SAM) format (*Li et al., 2009*), which is the standard output for most widely used mapping software, including BWA (*Li & Durbin, 2009*), Bowtie2 (*Langmead & Salzberg, 2012*), and CLC Genomics Workbench (http://www.clcbio.com). Subsequent to the generation of BAM files, a typical analysis of multiple metagenomic samples with anvi'o entails the following steps (Fig. 1): (1) generating a contigs database, (2) profiling each sample individually and merging the resulting single profiles, (3) visualizing results interactively, performing human-guided binning, or refining automatically identified bins, and (4) summarizing results.

## Contigs database

Anvi'o uses this essential database to store contig (or scaffold) information that does not vary from sample to sample (i.e., k-mer frequencies, functional annotation of open reading frames (ORFs), or GC content). To ensure that longer contigs are given more statistical weight during automated binning and more visibility in interactive displays, anvi'o breaks up large contigs into multiple *splits*, which remain soft-linked throughout the workflow and are reconstructed in the correct order in result summaries. The user can override the default split size of 20,000 bases when creating the contigs database. Smaller split sizes increase the resolution of information stored in databases and displayed in the interactive interface during later steps of analysis at the expense of added computational complexity and decreased performance for applications that require robust k-mer frequency statistics per split. When the user creates a contigs database from a given FASTA file, anvi'o identifies splits and computes k-mer frequency tables for each contig and split separately. Optionally, anvi'o can identify ORFs, process functional and taxonomic annotations for ORFs, and search contigs for hidden Markov model (HMM) profiles to be stored in the contigs database for later use. Currently, anvi'o installs four previously published HMM profiles for bacterial single-copy gene collections (*Alneberg et al., 2014*; *Campbell et al., 2013*; *Dupont et al., 2012*; *Creevey et al., 2011*). Presence or absence of these genes in contigs provides a metric for estimating the level of completeness of genome bins during the interactive human-guided binning (see 'Binning'). The system also generates completion and redundancy (multiple occurrence of one or more single-copy genes in a bin) statistics in real-time to inform human-guided binning. Beyond single-copy genes, users can populate the contigs database with curated HMM profiles to identify the presence of protein families of interest. The contigs database also stores inferred functions and likely taxonomic origin of all recognized ORFs. Users can provide these data as a standard matrix file or use one of the pre-existing parsers. The initial version supports annotation files generated by the RAST annotation server (*Aziz et al., 2008*), but the design allows inclusion of annotations from other sources.

## Profile database

In contrast to the contigs database, an anvi'o profile database stores sample-specific information about contigs. Profiling a BAM file with anvi'o creates a *single profile* that reports properties (i.e., the mean coverage) for each contig in a single sample. Each profile database links to a contigs database, and anvi'o can merge single profiles that link to the

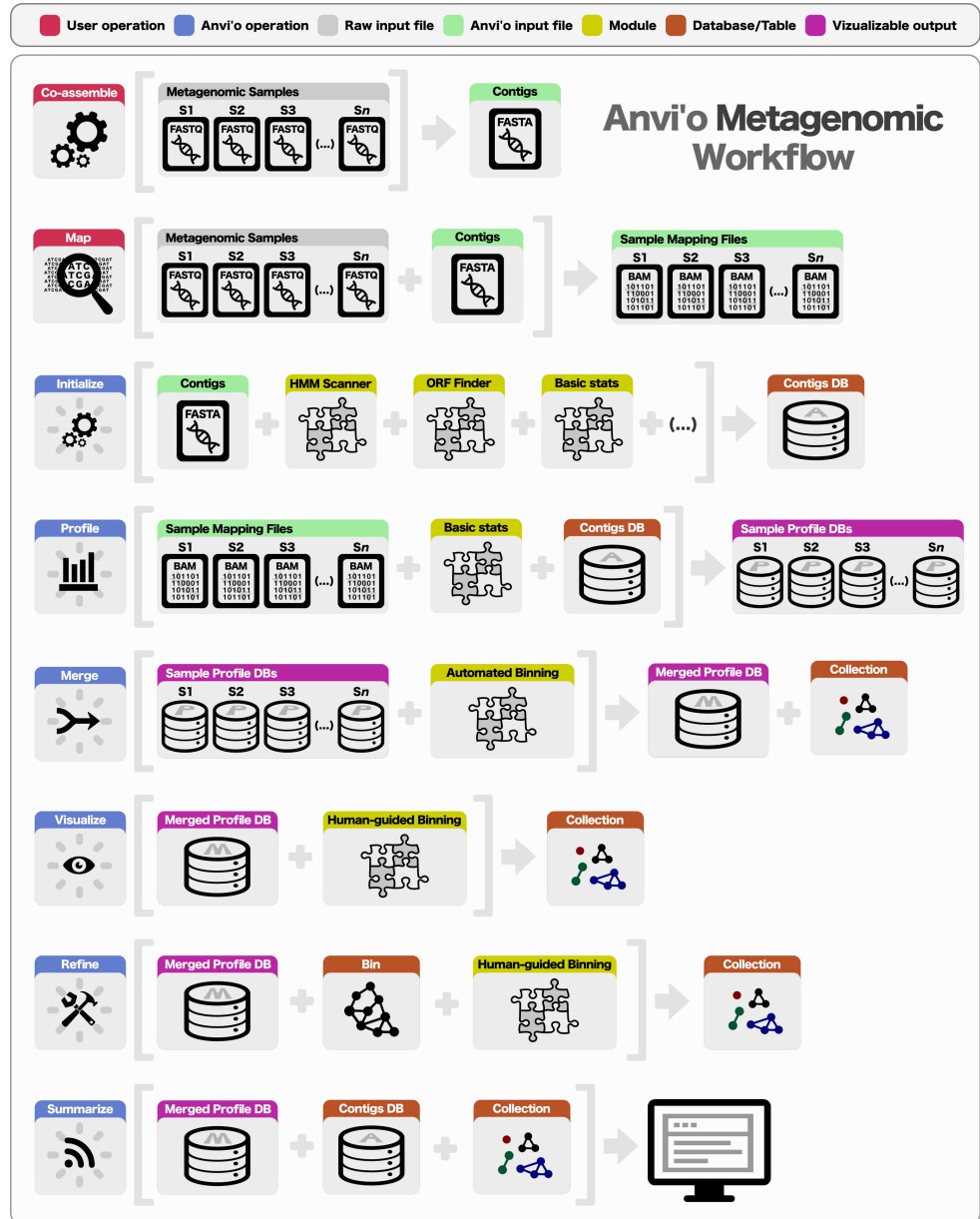

**Figure 1 Overview of the anvi'o metagenomic workflow.** Anvi'o can perform comprehensive analysis of BAM files following the initial steps of co-assembly and mapping. Initial processing of contigs and profiling each BAM file individually generate all the essential databases anvi'o uses throughout the downstream processing. Anvi'o can merge single profile databases, during which the unsupervised binning module would exploit the differential distribution patterns of contigs across samples to identify genome bins automatically, and store binning results as a collection. The optional visualization step gives the user the opportunity to interactively work with the data, and perform supervised binning with real-time completion and redundancy estimates based on the presence or absence of bacterial single-copy genes. The user can screen and refine genome bins, and split a single mixed genome bin into multiple bins with low redundancy estimates. Finally, the user can summarize collections that describe genome bins, which would create a static web site that would contain necessary information to review each genome bin, and to analyze their occurrence across samples.

same contigs database into *merged profiles*. The structure of single and merged profiles differs slightly: when multiple single profiles are merged, each property reported for each contig in single profiles becomes its own table in the merged profile database. For instance, the 'mean coverage column' from the single-profile data table for sample A and sample B would, when merged, become the 'mean coverage table' with sample A and sample B as columns. Anvi'o identifies these merged tables as *views*, and the user can switch between views in the interactive interface. This modularity fosters the quick implementation of new binning strategies and evaluation of results without requiring changes in the code. Profile databases also store other essential information such as frequencies of nucleotides at variable positions (see 'Computing variability'), and contig collections (see 'Binning').

### *De novo* characterization of nucleotide variation within samples

The alignment of short reads to a particular contig can generate one or more mismatches. The source of a mismatch may be artificial, such as stochastic sequencing or PCR error, however, some mismatches may represent ecologically informative variation. During the profiling step, anvi'o keeps track of nucleotide variation (base frequencies) among reads from each sample that map to the same community contig and stores that information in the profile database for each sample. To lessen the impact of sequencing and mapping errors in reported frequencies, anvi'o relies on the following conservative heuristic to determine whether to report the variation at a nucleotide position:

$$n_2/n_1 > \left(\frac{1}{b}\right)^{\left(x^{\frac{1}{b}} - m\right)} + c$$

where $n_1$ and $n_2$ represent the frequency of the most frequent and the second most frequent bases in a given nucleotide position, $x$ represents the coverage, and $b$, $m$, and $c$ represent empirically adjusted model parameters equal to 3, 1.45, and 0.05, respectively. This approach sets a dynamic baseline for the minimum amount of variation present at a given nucleotide position, as a function of coverage depth, for that nucleotide position to be reported. According to this conservative heuristic, the minimum ratio for $n_2$ to $n_1$ would be 0.29 for 20× coverage ($x$), 0.13 for 50× coverage, 0.08 for 100× coverage, and ∼0.05 for very large values of coverage as the minimum required ratio of $n_2$ to $n_1$ approaches $c$. This computation- and storage-efficient strategy reports a short list of sample-specific variable nucleotide positions that are unlikely to originate from PCR or sequencing errors. The user has the option to instruct the profiler to store all observed frequencies for more statistically appropriate but computationally intensive downstream analyses.

### Profiling variability

To interpret the ecological significance of sample-specific variable positions across samples, anvi'o installs a helper program, anvi-gen-variability-profile (AGVP). The user can specify filters that employ information from the experimental design to instruct AGVP's generation of a more refined variability profile. The current version of AGVP processes variable positions in a genome bin (see 'Genome binning') based on multiple

user-defined, optional filters, including the number of variable positions to sample from each split, minimum ratio of the competing nucleotides at a reported variable position, minimum number of samples in which a nucleotide position is reported as a variable position, minimum coverage of a given variable nucleotide position in all samples, and the minimum *scattering power* of a variable nucleotide position across samples. Samples in a merged profile can be organized into one or more groups ($g$) based on the nucleotide identity of the competing bases ($b$) at a given variable position, $p$. Scattering power then represents the number of samples in the second largest group. For example, at one extreme $b$ is identical in all samples at position $p$, so $g$ equals 1 and the scattering power of $p$ is 0. At the other extreme, $p$ harbors a different $b$ in every sample, thus $g$ is equal to the number of samples and the scattering power of $p$ equals 1. A value of $g$ between these two extremes yields a scattering power of $>1$. Since groups ($g$) are defined by not only the presence but also the identity of competing nucleotides at a given position across samples, the user can employ scattering power to query only those variable nucleotide positions that reoccur, and discard the ones that show stochastic behavior that is more likely to result from sequencing or PCR errors, or mapping inconsistencies.

## Genome binning

Anvi'o metagenomic workflow offers two modes for binning contigs into draft genomes: automated binning, and human-guided binning. The result of a binning process corresponds to a *collection* in a profile database. Each collection consists of one or more bins, with each bin containing one or more splits. When anvi'o merges multiple profiles, it passes coverage values of each split across samples to CONCOCT (*Alneberg et al., 2014*) for automated identification of genome bins. CONCOCT uses Gaussian mixture models to predict the cluster membership of each contig while automatically determining the optimal number of clusters in the data through a variational Bayesian approach (*Alneberg et al., 2014*). The merged profile database stores the result of automated binning as a collection. Anvi'o provides the user with a straightforward interactive interface to visualize automated binning results and to refine poorly resolved bins. CONCOCT is automatically installed with anvi'o, but the user can import clustering results from other automated binning techniques into separate collections in the profile database. During the merging step, anvi'o can generate a hierarchical clustering of contigs using multiple *clustering configurations*. A clustering configuration text file describes one or more data sources for the hierarchical clustering algorithm. A clustering configuration can request the retrieval of data for each contig from a profile database (such as a single attribute or a view), from a contigs database or from an external user-selected data source. A clustering configuration can also specify normalizations for each data source for anvi'o to employ when mixing multiple sources of information prior to the clustering analysis. The current version of anvi'o uses three default clustering configurations for merged profiles: 'tnf', 'tnf-cov', and 'cov'. Configuration 'tnf' uses k-mer frequencies to represent the sequence composition of contigs for clustering. The default 'k' is 4, but the user can set different values for 'k' in new contigs databases. Configuration 'tnf-cov' mixes k-mer frequencies from the contigs database with log-normalized coverage vectors from the merged profile database. This

configuration considers both sequence composition and the coverage across samples in a manner similar to CONCOCT. Configuration 'cov' uses only the coverage information from the profile database and ignores sequence composition. Each clustering configuration stores a Newick-formatted tree description of contigs in the profile database, which later becomes the central organizing framework of the interactive interface. Different clustering configurations can generate alternative organizations of contigs and the user can switch between visualizations of these organizations while working with the interactive interface to investigate different aspects of the data. The modular design behind the clustering infrastructure allows the user to add new clustering configurations without changing the code base and improves the human-guided binning process. Anvi'o can generate a complete and comprehensive summary of a collection upon completion of the binning process. The summary output is a user-friendly static HTML web site that can be viewed on any computer with or without an anvi'o installation or network access.

## Interactive interface

The interface has the ability to display large tree structures and overlay numerical and categorical data across the tree. This approach allows anvi'o to display splits with a particular organization dictated by a tree structure, and associate each leaf with a single item in each layer mapped across the entire tree. These items can display numerical or categorical information (such as GC-content, or taxonomy). The interface can direct human-guided binning and refinement of bins. The user can create a new collection to organize contigs into bins through mouse clicks, or load and modify collections previously stored in the profile database. The advanced search function of the interface can identify contigs that meet specific criteria and highlight their location on the tree, bin them together, or direct their removal from existing bins. The right-click menu provides fast access to NCBI tools to query public databases, and gives access to detailed inspection page for a given contig. The detailed inspection page displays coverage values and frequencies of variable bases for each nucleotide position in each sample for a given contig and it overlays ORFs and HMM hits on the contig. The interactive interface uses SVG objects for visualization and displayed trees can be exported as high-quality, publication-ready figures.

## Limitations

Certain steps of the anvi'o metagenomic workflow (such as profiling and merging) require intensive computation while others (such as visualization and human-guided genome binning) perform more efficiently on personal computers due to their interactive nature. Anvi'o optimally runs on server systems for non-interactive and parallelizable steps and on personal computers for visualization tasks. However, the design of anvi'o does not impose any limits on different configurations: the entire workflow can be run on a server as an independent web service, or on a personal computer with or without network access. The interactive interface can display a very large number of SVG objects, and its performance depends on the user's configuration since all interactive computations are done on the user's web browser. For the analyses in this study, we used cluster nodes with 48 to 512 Gb memory and 2.4 to 2.7 GHz CPUs to complete all computation-intensive

anvi'o tasks (i.e., profiling and sample merging) and a high-end laptop computer with 16 Gb memory and a 2.7 GHz CPU for all other anvi'o tasks (i.e., visualization and summary of results). We successfully used the interactive interface to visualize up to 500,000 SVG objects and trees that contained up to 25,000 leaves on our high-end laptop computer, however large visualization tasks decrease the responsiveness of the interface. One of the biggest limitations of anvi'o is the number of splits that can be clustered for human-guided binning. Human-guided binning may not be possible for datasets containing more than 25,000 splits because hierarchical clustering algorithms do not scale well with a time complexity of $O(n^2)$ or more. To work around this limitation, the user can mix automated and human-guided approaches by starting with automated clustering, and refining coarse genome bins through the 'anvi-refine' program. In this workflow, the user refines automatically identified bins with high redundancy estimations into high-quality draft genomes. The URL http://merenlab.org/projects/anvio provides a detailed guide for best practices.

## Preparation of publicly available sequencing datasets

### Noise filtering, assembly, mapping, and functional characterization of contigs

For each dataset, we analyzed the raw metagenomic data with illumina-utils library (*Eren et al., 2013*) version 1.4.1 (available from https://github.com/meren/illumina-utils) to remove noisy sequences using 'iu-filter-quality-minoche' program with default parameters, which implements the noise filtering described by *Minoche, Dohm & Himmelbauer (2011)*. CLC Genomics Workbench (version 6) (http://www.clcbio.com) performed all assembly and mapping tasks on a server computer with 1 TB memory and 4 CPUs (2.0 GHz each with ten cores) running Linux CentOS version 6.4. We used the default CLC parameters for assembly. For mapping, we required 97% sequence identity over 100% of the read length, and exported results as BAM files. We used RAST (*Aziz et al., 2008*) and myRAST (available from http://blog.theseed.org/downloads/) for functional characterization of contigs.

### Infant gut metagenomes

*Sharon et al. (2013)* collected daily infant gut samples at days 15–19 and 22–24 after birth including biological replicate samples on days 15, 17 and 22. Shotgun metagenomic analyses for the 11 samples share the NCBI Sequence Read Archive accession ID SRA052203. We co-assembled all samples after quality filtering. Since the reliability of k-mer frequency statistics and annotation specificity deteriorates with decreasing contig length, we chose an arbitrary contig minimum length of 1,000 base pairs. We mapped short reads from each sample back to these contigs (Table S1), then used anvi'o to perform profiling and merging of samples, followed by human-guided binning. After splitting draft genomes from our human-guided binning into 1,000 bp pieces, we used blastn version 2.2.28+ (*Altschul et al., 1990*) to determine their level of concordance with the draft genomes published by Sharon et al. (available at http://ggkbase.berkeley.edu/carrol). Analyses of variability between closely related draft genomes included only a single shotgun metagenome for each sampling day (using the metagenome with the largest number of reads from days 15, 17 and 22) to simplify computational complexity.

We used AGVP to access the variable positions reported in the merged profile database by specifying a maximum of 5 nucleotide positions from each split, and only retaining positions with a scattering power of three (see 'Profiling variabilty' for the definition). We used the interactive interface for human-guided genome binning.

### Deep Horizon samples

We used anvi'o to interrogate several previously published cultivar and single cell genomic, metagenomic, and metatranscriptomic datasets for environmental nucleic acid preparations from Pensacola Beach (Florida, USA) sand samples and Gulf of Mexico (GOM) water samples before and after the 2010 Deep Horizon oil spill.

### Overholt isolates

Data for ten culture genomes from *Overholt et al. (2013)* are publicly available as NCBI BioProject PRJNA217943. We concatenated all 10 cultivar genomes into a single FASTA file for downstream analyses.

### Rodriguez-R metagenomes

Raw metagenomic sequencing data for 16 samples from *Rodriguez-R et al. (2015)* are publicly available as NCBI BioProject PRJNA260285. After noise filtering, we mapped short reads from each sample back to Overholt isolates (Table S1). Anvi'o profiled and merged the resulting BAM files. In parallel, we co-assembled the metagenomic dataset, and discarded contigs smaller than 1,000 base pairs. After mapping short reads back to the co-assembled contigs (Table S1), anvi'o profiled individual BAM files and CONCOCT version 0.4.0 (*Alneberg et al., 2014*) performed automated binning. We summarized the CONCOCT results using 'anvi-summarize' and used 'anvi-refine' to interactively partition CONCOCT bins into high-quality draft genomes with high-completion and low-redundancy estimates.

### Mason single-cell genomes, metagenomes, and metatranscriptomes, and Yergeau metagenomes

The web site http://mason.eoas.fsu.edu/ posts quality-filtered data for three single-cell genomes (single amplified genomes; SAGs), three metagenomes, and two metatranscriptomes (*Mason et al., 2012*; *Mason et al., 2014*). We obtained quality-filtered data for metagenomes previously reported by *Yergeau et al. (2015)* from http://metagenomics.anl.gov/linkin.cgi?project=1012. We used the Yergeau metagenomic data only from the three samples collected from BM57 station, which is 3.87 km from the wellhead. Figure S1 summarizes our co-assembly, mapping, and analysis steps for these datasets. We first co-assembled short reads from the three Mason SAGs and independently co-assembled short reads from the three Mason metagenomes. Next, we mapped short reads from each of the Mason metagenomic, metatranscriptomic, and SAG datasets, as well as the three Yergeau metagenomes, to the co-assembled metagenomic dataset, and separately to the co-assembled SAG genome dataset generating two BAM files for each sample (Fig. S1; Table S1). We independently profiled each of the resultant BAM files (16 from Mason, 6 from Yergeau samples), and merged the 11 profiles from BAM file mappings to the metagenomic co-assemblies and separately merged the 11 profiles from

BAM file mappings to the SAG co-assemblies. We instructed anvi'o through an additional clustering configuration to employ only three Mason metagenomes for hierarchical clustering of contigs. We subsequently processed the merged profiles (1) to quantify the presence of short reads from metagenomic and metatranscriptomic reads matching to SAGs, (2) to quantify the presence of short reads from SAGs in the metagenomic contigs, and (3) to identify draft genomes through human-guided binning. To compare variability across samples, we generated variability profiles with AGVP program for each genome bin that we identified in the metagenomic assembly. To generate variability profiles for each genome bin, we instructed AGVP to sample up to 5 co-occurring variable nucleotide positions from each split in proximal and distal samples.

We used R version 3.1.2 (*R Development Core Team , 2008*) for the analysis of variance (ANOVA) (via 'aov' function) and to run the Tukey-Kramer post-hoc test on ANOVA results (via 'TukeyHSD' function), the R library ggplot version 1.0.0 (*Ginestet, 2011*) for all visualizations that were not done by anvi'o, and Inkscape version 0.48 (https://inkscape.org/) to finalize figures for publication. https://github.com/meren/anvio-methods-paper-analyses gives access to the shell and R scripts we implemented to generate variability profiles and to visualize results.

## RESULTS AND DISCUSSION

### Characterization of variable nucleotide positions in genome bins

The co-assembly of 11 samples in the infant gut dataset yielded 4,189 contigs with a minimal length of 1,000 bp, a total assembly size of 35.8 Mbp and an N50 of 36.4 kbp. On average, 92.4% (std: 4.43%) of all reads mapped back to contigs from each sample. The human-guided binning of the infant gut data with anvi'o converged upon 12 bacterial and one fungal genome bin that largely agree with the draft genomes *Sharon et al. (2013)* reported. Table S1 reports the quality filtering and mapping statistics, as well as the attributes of recovered genome bins. Figure 2 demonstrates the interactive interface of anvi'o, as it displays (1) the clustering dendrogram for contigs based upon their composition and differential coverage, (2) auxiliary layers that report information about contigs stored in the contigs database (GC-content, RAST taxonomy, number of genes, etc.), (3) view layers that report information about contigs across samples stored in the profile database (Panel A shows the mean coverage view, panel B exemplifies three other views), and (4) our draft genome bins. Having access to sample-independent auxiliary layers as well as sample-specific view layers that provide information for each contig in one interactive display improves the user's ability to work interactively with a given co-assembly. The URL http://merenlab.org/data/ gives read-only access to the interactive interface shown in Fig. 2 and the automatically generated anvi'o summary for this analysis.

Anvi'o can characterize positional nucleotide variation during the profiling step without requiring reference genomes. This information provides the basis for inferring subtle population dynamics within genome bins. We applied our analysis of nucleotide variation to three genome bins in the infant gut dataset: the two most abundant bins, *Enterococcus faecalis* and *Staphylococcus epidermidis*, with average coverage of ∼480×,

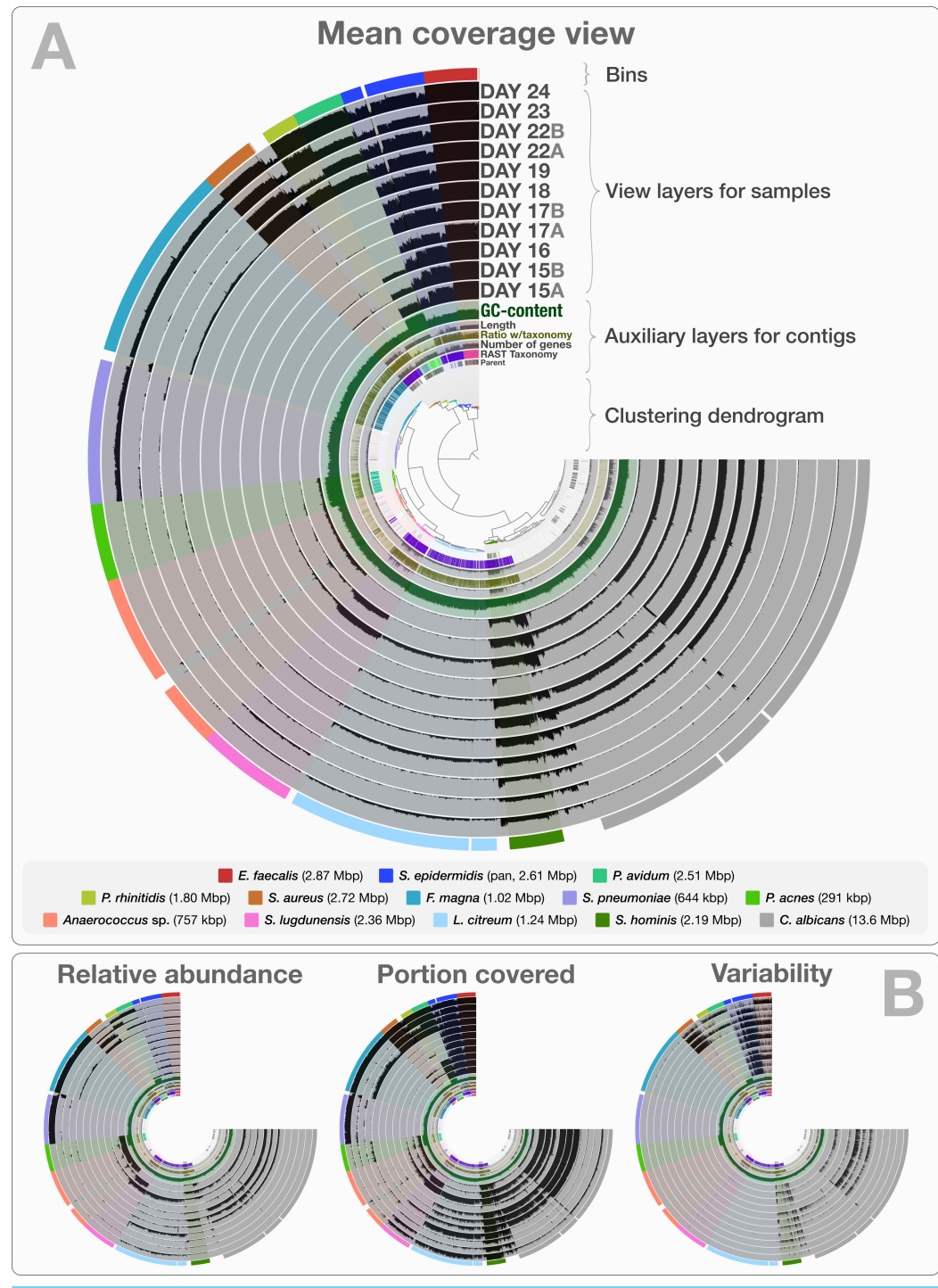

**Figure 2 Static images from the anvi'o interactive display for the infant gut dataset with genome bins.** The clustering dendrogram in the center of (A) displays the hierarchical clustering of contigs based on their sequence composition, and their distribution across samples. Each tip on this dendrogram represents a split (anvi'o divides a contig into multiple splits if it is longer than a certain amount of nucleotides, which is 20,000 bps in this example). Each auxiliary layer (continued on next page...)

**Figure 2 (...continued)**

represents essential information for each split that is independent of their distribution among samples. In this example auxiliary layers from the inside out include (1) the parent layer that marks splits originate from the same contigs with gray bars, (2) the RAST taxonomy layer that shows the consensus taxonomy for each open reading frame found in a given split, (3) the number of genes layer that shows the number of open reading frames identified in a given split, (4) the ratio with taxonomy layer that shows the proportion of the number of open reading frames with a taxonomical hit in a given split, (5) the length layer that shows the actual length of a given split, and finally (6) the GC-content layer. The view layers for samples follow the auxiliary layers section. In the view layers section each layer represents a sample, and each bar represents a datum computed for a given split in a given sample. (A) demonstrates the "mean coverage", where the datum for each bar is the average coverage of a given split in a given sample. (B) exemplifies three other views for the same display: "relative abundance", "portion covered", and "variability" of splits among samples.

and ∼60× respectively, as well as the *Staphylococcus aureus* bin that becomes abundant during the final three days of sampling with an average coverage of ∼50×. Anvi'o's profiling reported across all samples 3,241, 29,682 and 12,194 variable positions for the *E. faecalis*, *S. epidermidis*, and *S. aureus* bins respectively. Using the raw numbers for each sample in the three bins (Table S1), we first analyzed the *variation density*, which we define as the number of variable positions per kbp of contigs in a genome bin. *S. epidermidis* exhibited the highest variation density with a value of 2.27 on day 16 (second day of sampling). We then used AGVP to focus only on those nucleotide positions that showed consistent variation across samples by randomly sampling up to five nucleotide positions from each split. This analysis reported 418 positions for *E. faecalis*, 865 positions for *S. epidermidis*, and 158 positions for *S. aureus*. The *Staphylococcus* bins exhibited transition/transversion ratios of 2.21–2.67 consistent with expectations that transitions (mutations that occur from A to G, or T to C, and vice versa) usually occur more commonly than transversions (*Lawrence & Ochman, 1997*). In contrast, the *E. faecalis* bin displayed a transition/transversion ratio of 0.14. Our analysis also revealed very different nucleotide substitution patterns among the three groups. Increased variation density within contigs from the *E. faecalis* bins on even days alternates with lower variation density on odd numbered days (Fig. 3). The variation pattern, which includes conservation of nucleotide substitution patterns on alternate days at the same sites for *E. faecalis* bins suggests an underlying mechanism that does not affect other metrics such as coverage, and variation density. Initial inspection of this pattern suggests the possibility of 24-hour clonal sweeps that succumb to the re-establishment of a mixed population of a few different strains 24 h later. More likely, differences in methodology account for these patterns as Sharon et al. used two different size selections during the library preparation for these data: while they constructed libraries from odd-day samples with an insert size of 900 bp, they used an insert size of 400 bp for libraries from even-day samples. Variation in error frequencies between different Illumina sequencing runs or possibly differences in insert length that will affect cluster density might explain these patterns. Yet their non-random occurrence including clear patterns for each of the different major bins remains unexplained. In contrast, *S. epidermidis* and *S. aureus* bins did not show a bi-daily trend, and changes in their variability patterns did not follow the variability patterns anvi'o

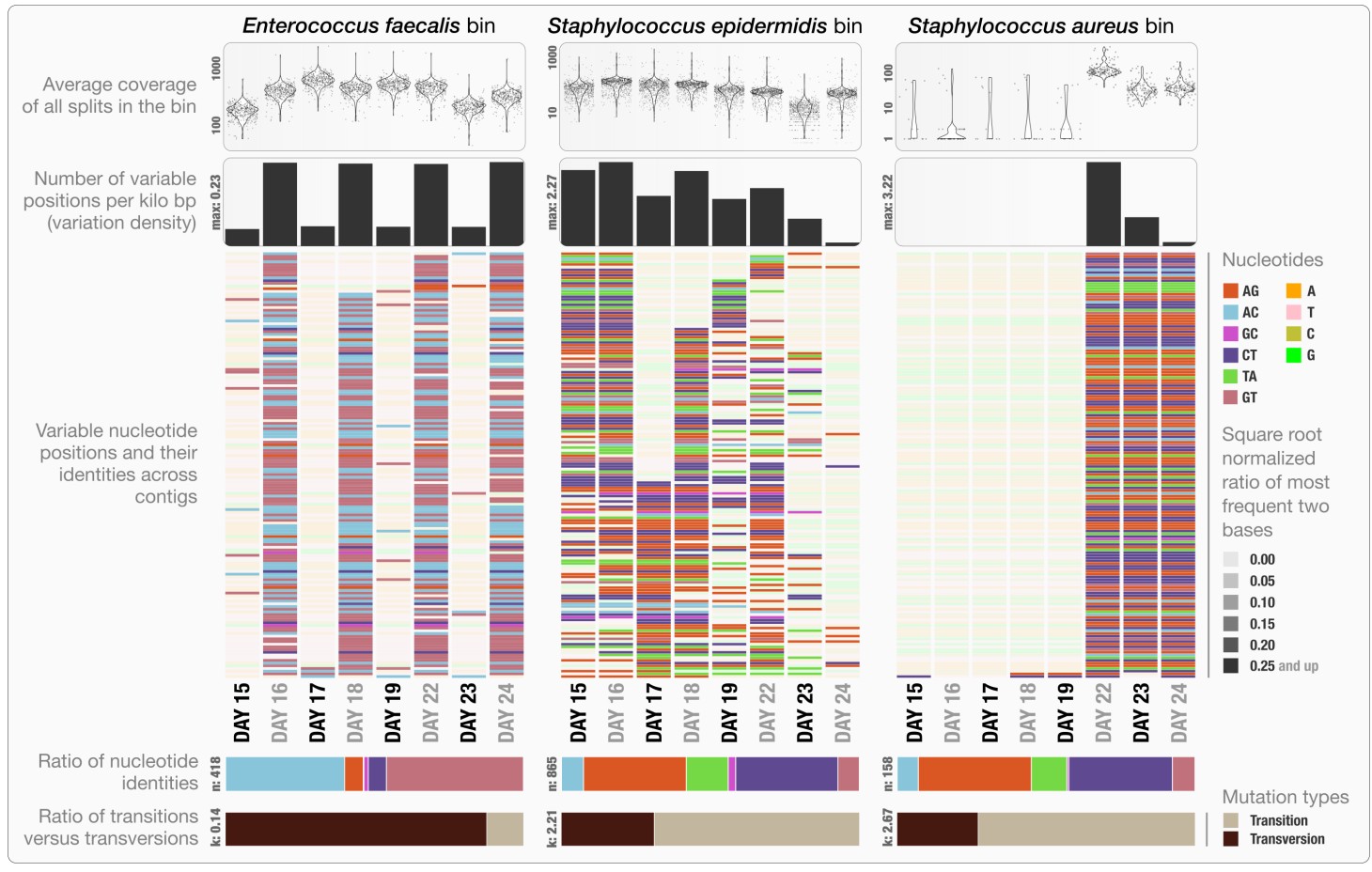

**Figure 3 Variable nucleotide positions in contigs for three draft genome bins.** The figure displays for each genome bin in each sample (from top to bottom), (1) average coverage values for all splits, (2) variation density (number of variable positions reported during the profiling step per kilo base pairs), (3) heatmap of variable nucleotide positions, (4) ratio of variable nucleotide identities, and finally (5) the ratio of transitions (mutations that occur from A to G, or T to C, and vice versa) versus transversions. In the heatmap, each row represents a unique variable nucleotide position, where the color of each tile represents the nucleotide identity, and the shade of each tile represents the square root-normalized ratio of the most frequent two bases at that position (i.e., the more variation in a nucleotide position, the less pale the tile is).

reported for *E. faecalis*. In their detailed analysis, Sharon et al. detected multiple strains in the *S. epidermidis* bin, members of which shifted throughout the sampling period. In our analysis, we detected a high variation density for the *S. epidermidis* bin, resonating with the highly mixed nature of this population. Variation density decreased in the *S. epidermidis* bin in time, and while the coverage of this bin did not change dramatically, the nucleotide variation nearly disappeared in samples from the last day (Fig. 3). This suggests a shift in the population with dominance by a relatively small number of *S. epidermidis* genomes. The absence of variability for *S. aureus* during the initial five-day sampling period reflects the mapping of very few metagenomic reads to these genomes, but by the 22nd day, *S. aureus* flourished with a very high variation density, which steadily decreased independent of the stable coverage.

Other investigators have utilized single bp changes to compare different variants of the same species based on reference genomes (*Zhang et al., 2006*; *Morelli et al., 2010*). While less frequent, identification of single bp changes has also been used to characterize heterogeneity in naturally occurring microbial populations through metagenomics (*Simmons et al., 2008*; *Morowitz et al., 2011*; *Tyson et al., 2004*). However, recovering detailed reports of single bp change patterns has not been straightforward due to the lack of adequate algorithms that can automatically identify and report nucleotide positions of high-variability inferred from multiple samples using contigs constructed *de novo* as reference for metagenomic short reads. The default metagenomic workflow of anvi'o now makes the under-exploited variability patterns accessible for every level of analysis. Application of our approach to draft genomes may lead to novel observations as well as more targeted investigations to describe underlying mechanisms that drive ecological processes. For instance, why does the *E. faecalis* population show bi-daily patterns in Sharon et al.'s dataset when *S. epidermidis* and *S. aureus* populations do not? Although exploring this question further falls outside the scope of our study, the observation of the single bp substitution patterns demonstrates the utility of anvi'o at providing deeper insights into metagenomic data.

## Holistic analysis of the microbial response to the Deep Water Horizon

In contrast to the infant gut dataset, the datasets related to the Deep Water Horizon oil spill represent a more challenging case given their size and complex nature. Following the DWH oil spill on April 20, 2010, investigators launched numerous molecular surveys to uncover bioindicators of oil pollution and to investigate the bioremediation capacity of indigenous bacteria. Multiple studies described the strong influence of oil on the bacterial community composition in the water plume, ocean sediments, and the shoreline, as well as enrichment of oil degradation genes in affected environments (*Hazen et al., 2010*; *Mason et al., 2012*; *Kimes et al., 2013*; *Mason et al., 2014*; *Kostka et al., 2011*; *Overholt et al., 2013*; *Rodriguez-R et al., 2015*). Our DWH collection included a metagenomic dataset generated by *Rodriguez-R et al. (2015)* from 16 sand samples collected from Pensacola Beach (Florida) during the three periods of beach oiling following the April 2010 DWH explosion: 'before' the oil had reached the shore, 'during' the oil contamination, and 'after' the oil was removed (Table S1). The dataset includes (1) four May 2010 samples collected before oil began to wash ashore the first week of June 2010, (2) four July 2010 and four October 2010 samples collected during the oiling event (the July and October samples each included one weathered sample with lower oil concentrations), and (3) four June 2011 samples collected after removal of oil from the beach. The original investigation of this dataset relied on taxonomic assignments of contigs from individually assembled samples without binning, and the authors observed a functional transition from generalist taxa during the oil pollution to specialists after the event. Our DWH collection also included genomes of 10 proteobacterial strains isolated from Pensacola Beach and Elmer's Island Beach (Louisiana) by *Overholt et al. (2013)* using samples collected in June and July 2010. In the original study, the authors suggested that these isolates represented the dominant oil degrading microbial populations by comparing their taxonomy to an independent

16S rRNA gene-based survey of the same environment (*Kostka et al., 2011*). The final dataset in our DWH collection included metagenome, metatranscriptome, and single-cell genome (SAG) data generated by *Mason et al. (2012)* and *Mason et al. (2014)* and *Yergeau et al. (2015)* from the oil spill water plume samples (Table S1). *Mason et al. (2012)* reported a rapid response of members of the Oceanospirillales to aliphatic hydrocarbons. *Yergeau et al. (2015)* investigated the same location one year after the event and detected *Oceanospirillales* in relatively low abundance. Our reanalysis of these data using anvi'o tests some of the previous assertions by providing contextual information and determining key genomic structures that were previously overlooked.

## Linking culture genomics to metagenomics

To estimate the abundance of Overholt isolates in the Pensacola Beach before, during, and after the oil contamination, we mapped the short reads from Rodriguez-R metagenomes to these 10 cultivar genomes. Overholt isolates recruited on average 0.00097% of the May 2010, 1.16% of the July 2010, 0.088% of the October 2010, and 0.0024% of the June 2011 metagenomic reads (Fig. 4 and Table S1). Anvi'o indicates high completion with little redundancy for these genomes (Table S1). Among the ten cultivars, *Alcanivorax* sp. P2S70 was the most frequently detected genome (Table S2). On average, the July 2010 metagenomes covered 96% of the *Alcanivorax* sp. P2S70 genome to ∼8× depth while the October 2010 metagenomes covered only 35% of the *Alcanivorax* sp. P2S70 genome with an average depth of ∼0.6×. Reads from the metagenome dataset of 452 million sequences mapped at very low levels to five of the isolates. Nonetheless, we observed a clear increase in the abundance of the ten genomes from 'before' to 'during' phases of the oil contamination, with a striking four thousand-fold increase of *Alcanivorax* sp. P2S70 between May and July 2010. The recovery of these genomes diminished in the two 'weathered' samples. Finally, the absence of short reads matching any of these ten genomes in samples from the 'after' phase, suggests that these isolates might depend on oil for their primary carbon source or that their growth might require syntrophic partnerships with other oil degrading microbes. The metagenomic data in our combined analyses support the hypothesis that increased oil concentration created a niche for the cultivars from Pensacola Beach. However, as these cultivars recruited only 0.0098% to 1.84% of the metagenomic reads from the same environment, our results also show that they were not the most abundant oil degraders (Fig. 4A) and contradict Overholt et al.'s 16S rRNA gene-based estimations (*Overholt et al., 2013*).

To access genomes of dominant oil degraders in the Gulf of Mexico shoreline without relying on cultivation, we co-assembled the Rodriguez-R dataset of 452 million reads. The *de novo* assembly yielded 56,804 contigs with a minimal length of 2.5 kbp and a total assembly size of 325.2 Mbp. The assembled bins recruited on average 20.4% of each sand metagenome during mapping (Table S1). Only 0.31% of the metagenomic reads were recruited to the cultivar genomes. The large size and fragmentation of the metagenomic assembly prevented us from a direct hierarchical clustering and visualization of all contigs for human-guided binning. Anvi'o offers a workflow for large datasets that

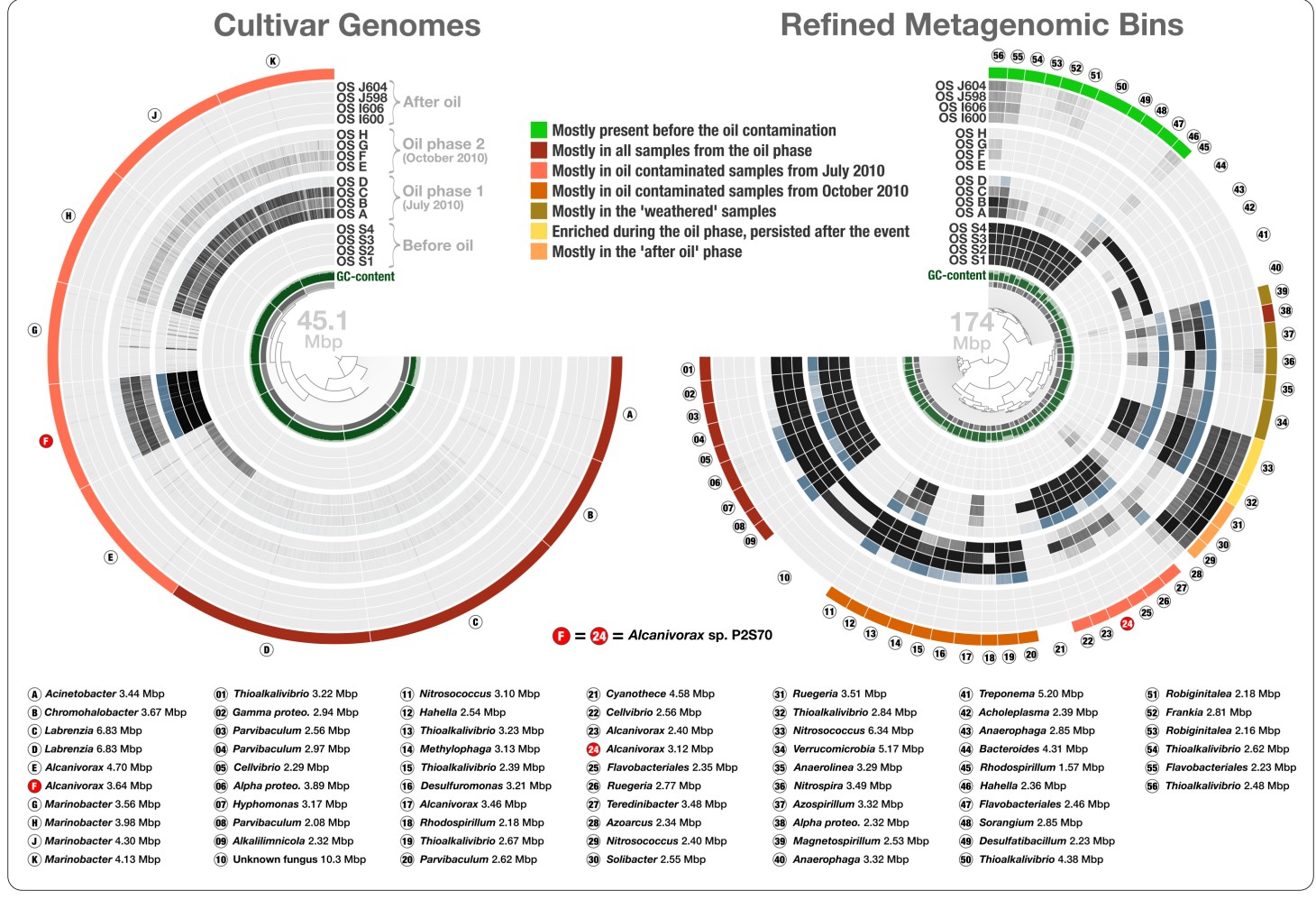

**Figure 4 Overholt culture isolates linked to the Rodriguez-R metagenomes of the beach sand microbial community.** The tree on the left displays the hierarchical clustering of 10 culture genomes based on sequence composition. Each view layer represents the "percent coverage" of each split in the Pensacola beach metagenomic dataset. The tree on the right displays the coverage-based hierarchical clustering of 56 environmental draft genomes we determined from the co-assembly of Pensacola Beach metagenomic dataset. The view layers display the "mean coverage" of each split in samples from the Pensacola beach metagenomic dataset. The most outer layer in both trees show the ecological pattern of a given genome bin during the period of sampling. Letters A to J identify culture genomes, and numbers 1 to 56 identify each metagenomic bin. The letter F, and the number 24, identifies two bins that represent the only genome that was present in both collections (*Alcanivorax* sp. P2S70). All genus- and higher-level taxonomy assignments are based on the best-hit function in RAST.

combines the automated and human-guided binning steps. CONCOCT's automated binning during anvi'o's merging step generated 81 bins with an average redundancy of 31.7%. We then visualized and manually partitioned these bins using anvi'o, creating 162 refined bins with an average redundancy of 1.96% (Table S2). In a more focused analysis, we used genome bins larger than 2 Mbp and/or more than 80% complete. The 56 draft genomes that fit these criteria had an average length of 3.11 Mbp (std: 1.31 Mbp) and their GC-content varied from 32.2% to 71.0%. We compared these draft genomes, along with the Overholt cultivars, to the closest matching reference genomes using the best-hit function implemented in RAST (Table S1). The RAST taxonomic inference supported

Overholt et al.'s assignments for 9 out of the 10 genomes derived from cultivation (our RAST analysis suggested the taxon name *Chromohalobacter* for Overholt et al.'s *Halomonas* PBN3 genome), and detected a total of 33 genera within the 56 draft genomes, which included a fungus (10.3 Mbp in length), and a Cyanobacterium affiliated with *Cyanothece* that harbors 60 genes encoding the photosynthesis aparatus. These taxonomic inferences largely agree with analysis of sample-centric contigs by *Rodriguez-R et al. (2015)*. The only organism present in both the Overholt cultivars and the draft genomes we identified in the Rodriguez-R metagenomes was *Alcanivorax* sp. P2S70. The metagenomic binning process recovered 86% of its genome ('bin 24' in Fig. 4). 95.9% of all proteins identified in this draft genome shared 99.2% protein identity with corresponding proteins identified in *Alcanivorax* sp. P2S70 genome (Table S2), and a total of 1,858 of them were identical between the two.

Seven of the 66 cultivar and draft genomes occurred primarily in a single sample. In addition, one draft genome was not characteristic of any phase (bin 28), and one draft genome represented a fungal organism (bin 10). The remaining 57 bacterial genomes exhibited one of seven distinct ecological patterns (Fig. 4 and Table S2): (1) mostly present before the oil contamination ($n = 11$), (2) characteristic of all samples from the oil phase ($n = 14$, includes 4 cultivars), (3) characteristic of oil contaminated samples from July 2010 ($n = 12$, includes the 6 remaining cultivars), (4) characteristic of oil contaminated samples from October 2010 ($n = 10$), (5) characteristic of the weathered samples ($n = 5$), (6) enriched during the oil phase and persisted after the event ($n = 2$), and finally (7) characteristic of the "recovered" phase ($n = 3$) (Fig. 4). Interestingly, the most frequently represented genus (*Thioalkalivibrio*, $n = 8$) occurred in four of the seven ecological patterns, emphasizing the importance of sensitive microbial population partitioning and the limitations of taxonomy-based binning. We grouped functions that occurred in our collection of bacterial draft genomes based on these seven ecological patterns. 2,621 of 12,982 functions occurred differentially across different ecological phases (ANOVA, Tukey-Kramer post-hoc test, $p < 0.05$; Table S2).

Genes involved in oil degradation and described by *Rodriguez-R et al. (2015)* likely drive shifts in the beach microbial community during oil spills. Oil-degrading microbes detected in beach sand might be members of the rare biosphere and/or originate from the ocean. Here we examined the functional annotation of genes in our bins for insight into the environmental origin of oil-degrading bacteria. Among the functions characteristic of genomes enriched during the oil phase were the acquisition and metabolism of urea (Table S2). Urea is a dissolved organic nitrogen compound that can occur at highly abundant levels in coastal oceanic systems and serves as a main source of nitrogen for marine bacteria (*Solomon et al., 2010*). The apparent lack of urea metabolism in genomes characteristic of the uncontaminated beach samples in this dataset suggest this compound does not serve as a primary source of nitrogen in the innate microbial populations. On the other hand, the acquisition of carbon sources through oil degradation processes likely triggers an increased need for micronutrients such as nitrogen, and urea might represent an important source of nitrogen to support the bioremediation process. Urea-related functional traits suggest

a lifestyle adapted to the marine environment, lending support to the hypothesis of an oceanic origin for microbes involved in the bioremediation process at the oil-contaminated Pensacola Beach.

Co-assembly of the metagenomic data, and the identification of draft genomes through anvi'o, revealed a more comprehensive perspective on community changes in response to the oil spill relative to the cultivars alone, which depicted only two ecological patterns and represented relatively low abundance populations. The most significant functional difference between the 10 cultivars and the 59 draft genomes involved the arsenic resistance protein ArsH ($p$: 4.01e–21), which occurred in all culture genomes, but in only one bacterial draft genome. While multiple factors likely affect the cultivability of microbes when using oil as a sole source of carbon, arsenic, a toxic consequence of most oil spills (*Cozzarelli et al., 2015*), might differentially impact the fitness of oil degraders and prevent the isolation of some of the most promising populations for bioremediation processes.

## Linking single-cell genomes, metatranscriptomes, and metagenomes

The Mason data (*Mason et al., 2012*; *Mason et al., 2014*) contained metagenomes of ocean water samples collected five weeks after the oil spill at three locations: 1.5 km from the wellhead ('proximal' sample), 11 km from the wellhead ('distal' sample), and 40 km from the wellhead ('uncontaminated' sample). In addition to the metagenomes, the authors generated metatranscriptomic data from the proximal and distal samples, and isolated three single-cell genomes (SAGs) from the proximal sample (Table S1). The Yergeau data (*Yergeau et al., 2015*) contained metagenomes of ocean water samples collected one year after the oil spill at multiple depths at two locations: 3.87 km (BM57) and 37.8 km (A6, control station outside the plume) from the wellhead (Table S1). Consistent with previous studies (*Hazen et al., 2010*; *Redmond & Valentine, 2012*), Mason et al.'s analysis suggested that the taxonomic group DWH *Oceanospirillales* dominated the bacterial community composition and activity within the oil plume. Furthermore, Mason et al. suggested, through their standalone analysis of SAGs, metagenomic, and metatranscriptomic datasets, that the dominant and active *Oceanospirillales* possessed genes encoding a near-complete cyclohexane degradation pathway. The multifaceted datasets from Mason et al.'s samples taken shortly after the event and Yergeau et al.'s later samples provide an opportunity to investigate the microbial response to the DWH oil spill in a comprehensive manner. Anvi'o facilitated a holistic analysis of this composite dataset by linking separate sources of data into one unified perspective that led to a high-resolution genomic analysis of the dominant DWH *Oceanospirillales* population in time and space.

The co-assembly of 46.8 million reads representing 3 SAGs yielded 941 contigs with a minimal length of 1 kbp, a total assembly size of 2.88 Mbp and an N50 score of 3.88 kbp. Clustering of contigs based on their sequence composition ($k = 4$) formed two distinct groups that represent genetic structures originating from *Colwellia* and *Oceanospirillales*, in agreement with Mason et al.'s findings (Fig. 5A). When combined, the two *Oceanospirillales* SAGs provided a draft genome of 1.91 Mbp that included ∼1.3 Mbp

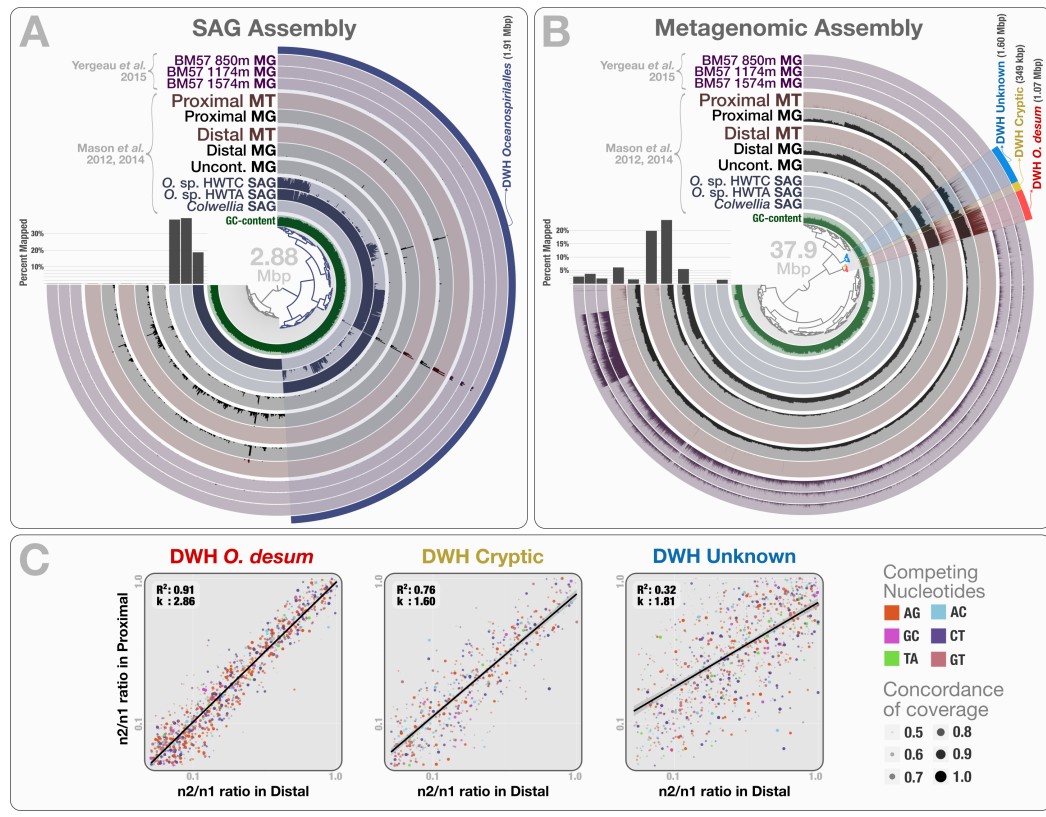

**Figure 5 Mapping of samples to SAGs and metagenomic assembly, and nucleotide frequencies and identities of variable positions in three bins.** (A) shows the mapping of *Mason et al. (2012)* and *Mason et al. (2014)* samples, as well as the three *Yergeau et al. (2015)* depth profiles collected from a location close to Mason et al.'s proximal station, to the co-assembly of the three SAGs. The dendrogram shows the sequence composition-based hierarchical clustering of the community contigs with the "portion covered" view, where each bar in the sample layers represents the percentage of coverage of a given contig by at least one short read in a given sample (i.e., if each nucleotide position in a contig is covered by at least one read, the bar is full). (B) shows the mapping of the same samples to the co-assembly of the three Mason et al. metagenomes. The dendrogram shows the sequence composition- and coverage-based hierarchical clustering of the community contigs with the "mean coverage" view, where each bar in the sample layers represents the average coverage of a given contig in a given sample. Bar charts on the left-side of dendrograms both in (A) and (B) show the percent mapped reads from each sample to the assembly. (C) compares the identity and frequency of the competing nucleotides at the co-occurring variable positions in three bins identified in the (B): DWH *O. desum*, DWH Cryptic, and DWH Unknown. *X*- and *Y*-axes in each of the three plots represent the ratio of the second most frequent base ($n_2$) in a variable position to the most frequent base ($n_1$) in distal, and proximal samples, respectively. Each dot on a plot represents a variable nucleotide position. The color of a given dot represents the identity of competing nucleotides. The size of a given dot increases if the coverage of it is similar in both samples, where size equals to '1—std(coverage in proximal, coverage in distal)'. Linear regression lines show the correlation between the base frequencies at variable nucleotide positions. Each plot also displays the $R^2$ values for linear regressions, and the ratio of transition versus transversion rates ($k$).

of shared contigs with a sequence identity over 99%. However, only 0.16–0.64% of the metagenomic and metatranscriptomic reads mapped to the *Oceanospirillales* SAGs which indicates low levels of relative abundance (Table S1). Moreover, a majority of mapped reads represented non-specific regions of ribosomal RNA operons (Fig. 5A). These results

disagree with previous findings, and suggest that the recovered SAGs do not represent the dominant or active members of the microbial community at the time of sampling. Why did none of the three single-cell captured organisms represent an abundant member of the microbial community? This incongruence may reflect a methodological bias, where the population structure of captured single cells diverges from the rank abundance curve of the organisms that occur in the sampled environment.

To recover the draft genome of DWH *Oceanospirillales* population, we co-assembled the metagenomic dataset of Mason et al. (397.9 million reads), which yielded 19,954 contigs longer than 1 kbp (N50: 1.88 kbp), with a total length of 37.9 Mbp. These contigs recruited reads corresponding to 5.83% to 23.6% of the Mason metagenomes, 1.52% to 3.58% of the Yergeau metagenomes and 1.58% to 6.12% of the Mason metatranscriptomes during the mapping (Table S1). Clustering of contigs by sequence composition and coverage patterns across the three Mason metagenomes revealed a distinct bin that contained 1.07 Mbp with a completion score of 62.8%. Here we temporarily name this bin as "DWH *Oceanospirillales desum*" to avoid confusion with the DWH *Oceanospirillales* previously identified through SAGs. DWH *O. desum* recruited 77.8% and 79.5% of all mapped metagenomic reads in the proximal and distal samples, respectively. In contrast, DWH *O. desum* recruited only 3.55% of mapped reads in the uncontaminated sample, emphasizing the dramatic shift in its abundance between uncontaminated and contaminated samples five weeks after the oil spill (Fig. 5B). Furthermore, only 0.08% to 0.98% of mapped reads from the Yergeau metagenomes were recruited by DWH *O. desum*, indicating that the abundance of this microbial population was not only limited in space, but also in time. The result also suggests that the so-called "uncontaminated station" from Mason et al. might have been already tainted with oil at the time of sampling, as the relative abundance of DWH *O. desum* was >20 fold higher in the corresponding metagenome compared to its average in the six Yergeau metagenomes.

DWH *O. desum* recruited 97% and 99% of the mapped metatranscriptomic reads from the distal and proximal samples from Mason et al., respectively. Since we had not used the metatranscriptomic data for clustering, the extensive mapping of the transcriptome reads to DWH *O. desum* confirms the link between its abundance in this dataset and its activity in the environment. The 1,375 nt 16S rRNA gene from DWH *O. desum* matched the uncultured *Oceanospirillales* bacterium clones from proximal and distal stations published by *Hazen et al. (2010)* with over 99% sequence identity. The first cultured organism matched to *O. desum* 16s rRNA by BLAST against the NCBI's refseq_genomic database was *Oleispira antarctica* strain RB-8 (Oceanospirillales; Oceanospirillaceae) at 92% identity, and the *O. desum* 23S rRNA gene matched that of *Oleispira antarctica* at 93% identity. These results indicate that DWH *O. desum* represents the abundant and active *Oceanospirillales* population in the environment at the time of sampling. We also analyzed the variable positions that occurred in DWH *O. desum* population in proximal, distal, and uncontaminated samples. Despite the high variation density across samples, frequencies of the competing bases at positions of high nucleotide variation for DWH *O. desum* were nearly identical in proximal, and distal samples, indicating a similar population structure for DWH

*O. desum* at both sampling stations (Fig. 5C). Our analysis of the metatranscriptomic data that mapped to the DWH *O. desum* bin revealed the expression of genes regulating the synthesis and export of lipids (lipid-A-disaccharide synthase, lipid A export), lipoproteins (protein LolC) and capsular polysaccharides (proteins LptB, KpsD, KpsE, KpsM and KpsT), known to act as bio-surfactants in oil degrading bacterial models by increasing the solubility of hydrocarbons (*Ron & Rosenberg, 2002*). Aside from the ribosomal machinery, one of the most highly expressed genes coded for a cold-shock protein, which might aid the metabolism of this psychrophilic population in a temperature suboptimal for their growth. Overall, the functional activity of DWH *O. desum* exhibits activity consistent with known oil degradation mechanisms coupled with a state of cellular stress.

We identified two bins adjacent to DWH *O. desum* that were strongly enriched in proximal and distal samples compared to the uncontaminated station and samples collected one year after the event. These clusters showed remarkable activity and coverage that were distinct from DWH *O. desum* and from each other (Table S3). One of these two clusters is the size of a small bacterial genome (~1.6 Mbp). However, we found no single-copy gene markers; hence, a puzzling completion level of 0%. We refer to this cluster as "DWH Unknown". The second bin had a total length of only 0.35 Mbp, and we refer to it as "DWH Cryptic". We performed an analysis of polymorphism on these bins to compare the populations they represent in distal and proximal samples. Our examination indicated that the frequencies of bases at variable positions showed much less agreement compared to DWH *O. desum* between proximal and distal samples. This observation may indicate a subtle change in the population structure between the two stations. Alternatively, it may merely reflect technical limitations, since the coverage of both bins by data from distal station samples was much lower than that of DWH *O. desum*. Figure S2 demonstrates the change in coverage of the reported variable nucleotide positions in three contigs that represent each genome bin. The overall functional profiles of these two clusters did not resemble a typical bacterial genome: while the genes encoding for the ribosomal machinery were largely missing, pathways for phage machinery and protection against phages (CRISPRs and the type I restriction-modification system) were dramatically enriched (Table S3). In the case of DWH Unknown, most expressed genes encoded proteins involved in the synthesis, transport, and export of capsular polysaccharides. The most highly expressed gene in DWH Cryptic encoded cytochrome P450 hydroxylase, an enzyme involved in the metabolism of hydrocarbon (*Ortiz de Montellano, 2010*). Other highly expressed genes were associated with the transport and export of capsular polysaccharides, as well as CRISPR-associated proteins. These bins likely represent phages or plasmids. We did not detect any genes related to ribosomal machinery in these bins despite their rather large size, therefore their presence in the environment would be missed by 16S rRNA gene-based surveys, as well as metagenomic analyses that do not perform genome binning. Their enrichment in the polluted stations and metabolic activity centered on polysaccharide synthesis and export suggests a role in hydrocarbon degradation, yet the origin of these two genetic structures remains unclear. The anvi'o summary of the three bins is available at: http://merenlab.org/data/.

## Anvi'o as a community platform

The ability to interact with metagenomic and metatranscriptomic data, identify and refine draft genome bins with real-time feedback, and report final results in a comprehensive and reproducible manner are essential needs for the rapidly growing field of metagenomics. Anvi'o introduces a high-level, dynamic visualization framework to better guide 'omics analyses and to communicate results, while it empowers its users with easy-to-use interfaces that require minimal bioinformatics skills to operate. Because of its modular structure, anvi'o can mix information the profiling step generates from the raw input files with additional user-provided information in a seamless manner (i.e., external human-guided or automated binning results, experimental organization of contigs, views, or simply additional data or metadata layers). Through this flexibility, anvi'o does not impose specific analysis practices, and encourages question-driven exploration of data.

Anvi'o is an open source project, and it welcomes developers. By abstracting the monotonous steps of characterizing and profiling metagenomic data, the platform gives its users with programming skills the ability to access internal data structures and implement novel ideas quickly. For example, anvi'o profiler computes several standard properties for each contig (i.e., mean coverage, and variation density), however, it can accommodate new attributes produced by any algorithm that yields a numerical value for a given contig. The addition of a new experimental property by an experienced user would automatically integrate into the workflow, resulting in a new view in the interactive interface and becoming accessible to clustering configurations for enhanced human-guided binning and visualization immediately. We developed anvi'o using modern programming languages and paradigms, relied on easy-to-query and self-contained database files for data storage, and used open technologies for visualization tasks. These properties leverage anvi'o as a community platform that can support the development, testing, and dissemination of new approaches.

## CONCLUSIONS

Anvi'o is an open-source, extensible software platform built upon open technologies and standard file formats to study 'omics data. In this study we used anvi'o to combine environmentally linked datasets of different types from multiple investigators, to identify draft genomes in both human-guided and automated manners, to infer population dynamics within draft genome bins through *de novo* characterization of nucleotide variation, to visualize layered data and generate publication-ready figures, and to summarize our findings. Through anvi'o we identified systematic emergence of nucleotide variation in an abundant draft genome bin in an infant's gut, and extended our understanding of the microbial response to the 2010 Deepwater Horizon Oil Spill. Anvi'o's ability to integrate, analyze, and display data of diverse origins empowers its users to fully explore their sequencing datasets in order to address a wide variety of questions.

## ACKNOWLEDGEMENTS

We thank Faruk Uzun, Doğan Can Kilment, Gökmen Göksel, S. Çağlar Onur, and Gökmen Görgen for their contributions to the code base, and Rich Fox for administering our

servers. We thank Inés Martínez for testing anvi'o, and for her valuable suggestions throughout the development of the platform. We thank Sheri Simmons for suggesting the application of oligotyping to the metagenomic data to characterize single-nucleotide variation. We also thank Itai Sharon, Luis M. Rodridugez-R, Will A. Overholt, Olivia U. Mason, Etienne Yergeau, and their colleagues for making valuable datasets available to the science community and for answering our questions.

### Funding

AME was supported by the G. Unger Vetlesen Foundation. The project was supported by the Frank R. Lillie Research Innovation Award given by the University of Chicago and the Marine Biological Laboratory. The funders had no role in study design, data collection and analysis, decision to publish, or preparation of the manuscript.

### Grant Disclosures

The following grant information was disclosed by the authors:
G. Unger Vetlesen Foundation.
Frank R. Lillie Research Innovation Award.

### Competing Interests

The authors declare there are no competing interests.

### Author Contributions

- A. Murat Eren and Tom O. Delmont conceived and designed the experiments, performed the experiments, analyzed the data, contributed reagents/materials/analysis tools, wrote the paper, prepared figures and/or tables, reviewed drafts of the paper.
- Özcan C. Esen performed the experiments, contributed reagents/materials/analysis tools, reviewed drafts of the paper.
- Christopher Quince contributed reagents/materials/analysis tools, reviewed drafts of the paper, developed the CONCOCT module for the platform.
- Joseph H. Vineis contributed reagents/materials/analysis tools, reviewed drafts of the paper.
- Hilary G. Morrison and Mitchell L. Sogin contributed reagents/materials/analysis tools, wrote the paper, reviewed drafts of the paper.

### Data Availability

Meren Lab (http://merenlab.org/data/) provides access to supporting data, live demonstrations, summaries, and the code to regenerate figures.

### Supplemental Information

Supplemental information for this article can be found online at http://dx.doi.org/10.7717/peerj.1319#supplemental-information.

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
