# Peer review of "Anvi’o: an advanced analysis and visualization platform for ‘omics data"

_PeerJ, doi:10.7717/peerj.1319_

## Round 0.1 · original submission · Minor Revisions

As mentioned below Peerj does not offer copyediting. One of the reviewers noted some small errors in the English language. Please consider having the manuscript text checked by a native English speaker.

Reviewer 1 ·

Basic reporting

The paper is mostly well written and provides generous detail on methods and approach. As a result, it is quite long and readability could be improved by moving details provided in the Materials & methods and Results section to supplementary sections - however, I do not consider this a requirement.

Some minor issues:
- the abstract and introduction are vague in some points for those that have not read the full paper (e.g. "Using 'omic techniques [...] requires [...] incorporate subtle differences that enable greater resolution of complex data").
- the reason for splitting and considerations for choosing split size are not clear from the text.
-the discussion on the threshold chosen for reporting nucleotide variation is confusing: (a) what is the source for the formula for y or, if there is none, what is the underlying reasoning? (b) what is the motivation for the parameter choices for b, m and c?
- the discussion on scattering power is unclear: (a) the extremes are 0 (no variation) and 1 (each sample a different base), but situations between these extremes yield a scattering power > 1; wouldn't one expect a scattering power between 0 and 1? (b) it is not immediately obvious how this helps to select positions that vary "consistently across samples" (i.e., what is "consistency"?). Wouldn't a better measure be the size of the 2nd largest group divided by that of the largest group - 0 when there is just 1 group, 1 when all groups are equally sized, between 0 and 1 otherwise?
- the paper contains a number of small spelling problems, mainly mixing singular and plural.

Experimental design

As this manuscript reports on a computational tool rather than novel experimentation, this is less applicable. Nevertheless, while I am not an expert in the application areas, I believe the authors process and analyze the datasets they obtained from literature in a rigorous manner.

Validity of the findings

See the comments on "Experimental Design".

Additional comments

Anvi'o seems to be an extensive piece of work and a worthwhile addition to the toolbox of those studying metagenomics. The graphical quality and ease-of-use are impressive. Installation and use are not very straightforward, but that is to be expected for a project of this scope.

Reviewer 2 ·

Basic reporting

No Comments.

Experimental design

Materials & Methods:
line 307: Authors note using CLC workbench for assembly, but need to note the parameters used for co-assembly (default?). Also, the authors do not provide details on the type of machine that was required for co-assembly, this could help orient the reader to requirements for analyses prior to using the Anvi-o software.

line 316: How did the authors decide on the 1000 bp limit for contigs? Was this an arbitrary cutoff to increase computational efficiency based on the datasets examined?

line 365: were only three metagenomes used from the Mason dataset to reduce total computational complexity?

Results:
line 566: statistical methods are missing from the materials and methods section.

Validity of the findings

No comments.

Additional comments

The authors present an exciting new software package called Anvi’o that combines user generated data from co-assemblies, read mapping, and annotation with a powerful workflow and visualization interface that can be run from a user’s laptop. The workflow starts from BAM files and consists of steps to (i) profile data based on sequence properties and annotations, (ii) organize the data into genome bins, and (iii) examine variation (at the SNP level) to define strain level differences based on contextual (metadata) information defined by the user. The workflow also contains features to allow the user to refine their analyses and visualize output to create publication quality figures.
The software is open source and follows best practices for software design through the use of github. Perhaps even more laudable, is the author’s careful software design using technologies, such as SQLite databases and optimized java code, that are easily portable across computational platforms to reduce the overhead needed to run the workflow. As a reviewer, I was impressed by the ease in downloading and installing the Anvi’o code on both my personal laptop and our high performance compute cluster to perform my own analyses and visualizations. Moreover, the flexibility of providing my own annotations (taxonomic and functional) plus other evidence such as k-mer frequencies was commendable.

The manuscript presents a re-analysis of prior metagenomic work to demonstrate the utility of the software in analyzing data from diverse datasets. The first data set is derived from a infant gut microbiome analysis as the infant matures and demonstrates the utility of Anvi’o for temporal investigations of microbial communities. The second and larger dataset consists of cultivar and single cell genomes, metagenomes and metatranscriptomes from the 2010 deep horizon oil spill from water samples from the Gulf of Mexico and Pensacola Beach (Florida). Both examples demonstrate how Anvi’o enhances researchers’ capabilities to create genomic bins and find & visualize nucleotide variation among strains, thereby streamlining work that previously required time consuming analysis by a skilled bioinformatician. Moreover, the authors uncover intriguing patterns in the data sets that were not uncovered by the original authors. For example population level differences in species bins were found in the case of the infant gut microbiome that are not consistent with nucleotide substitution patterns in the case of E. Faecalis & species that were considered dominant in the 2010 deep horizon oil spill may in fact be less abundant. Each of these examples highlight the utility of Anvi’o in analyzing metagenomes based on sequence composition patterns, annotations, genomic bins and population level variation that will contribute to the quality of metagenomic analyses put forth using this software.

Limitations of the software stem from issues related to computational capacity on individual machines for visualization & data scale and hierarchical clustering algorithms common to all bioinformatics analyses. As such, the authors provide valuable strategies for inferring broader patterns in large-scale data sets by limiting analyses to contigs that are >1000bp and mixing supervised and unsupervised approaches to binning contigs into genome bins. Thus, a bonus of the paper are strategies for mining through large scale datasets using Anvi’o analyses.

The manuscript is well-written and associated code is well-documented and I recommend that the paper is published.

---

## Round 0.2 · accepted · Accept

I appreciate the careful revision of your manuscript based on the comments raised by the reviewers.